EMBO
reports

# Negative feedback via RSK modulates Erk-dependent progression from naïve pluripotency

Isabelle RE Nett[1,†,‡], Carla Mulas[1,†] (iD), Laurent Gatto[2,3], Kathryn S Lilley[2,4] & Austin Smith[1,4,*] (iD)

## Abstract

Mitogen-activated protein kinase (MAPK)/extracellular signal-regulated kinase (ERK) signalling is implicated in initiation of embryonic stem (ES) cell differentiation. The pathway is subject to complex feedback regulation. Here, we examined the ERK-responsive phospho-proteome in ES cells and identified the negative regulator RSK1 as a prominent target. We used CRISPR/Cas9 to create combinatorial mutations in *RSK* family genes. Genotypes that included homozygous null mutations in *Rps6ka1*, encoding RSK1, resulted in elevated ERK phosphorylation. These RSK-depleted ES cells exhibit altered kinetics of transition into differentiation, with accelerated downregulation of naïve pluripotency factors, precocious expression of transitional epiblast markers and early onset of lineage specification. We further show that chemical inhibition of RSK increases ERK phosphorylation and expedites ES cell transition without compromising multilineage potential. These findings demonstrate that the ERK activation profile influences the dynamics of pluripotency progression and highlight the role of signalling feedback in temporal control of cell state transitions.

**Keywords** embryonic stem cells; mitogen-activated protein kinase; pluripotency; RSK; signalling feedback

**Subject Categories** Post-translational Modifications, Proteolysis & Proteomics; Signal Transduction; Stem Cells

## Introduction

Mouse embryonic stem (ES) cells are pluripotent cell lines derived from the naïve epiblast of the pre-implantation embryo [1–5]. They can be maintained using two selective chemical inhibitors (2i) to reduce activity of glycogen synthase kinase-3 (GSK3) and to suppress entirely signalling through the ERK1/2 MAP kinase pathway [6]. The cytokine leukaemia inhibitory factor (LIF) can further stabilise naïve identity and enhance self-renewal [7–10]. In 2i and LIF (2iLIF), ES cells propagate as near-homogeneous populations of naïve stem cells. In this ground-state condition, ES cells retain a remarkable degree of phenotypic and functional similarity with the *in vivo* naïve epiblast [1,6,11–14].

Upon withdrawal from 2iLIF, ES cells enter the pathway to multilineage differentiation while continuing to proliferate [15–17]. This transition can occur in defined media without exogenous inductive signals, implying that it is intrinsically driven and that self-renewal entails active suppression of the effector pathways for developmental progression [18]. The individual 2iLIF components each reduce and delay differentiation but a pairwise combination is required for long-term self-renewal and all three are optimal [7,10]. The most important effect of partial inhibition of GSK3 is to abrogate the capacity of the transcriptional repressor Tcf3 (gene name *Tcf7l1*) to downregulate key components of the ES cell gene regulatory network [19–21]. Conversely, LIF acts by boosting expression of members of the network [22–24]. Genetic perturbations have established that ERK1/2 signalling downstream of fibroblast growth factor 4 (FGF) is important for timely and efficient exit from the naïve state and subsequent commitment [15,25–31]. Consistent with this, inhibition of either FGF receptor or the MEK/ERK1/2 pathway markedly impedes ES cell differentiation [6,32,33]. Circumstantial data suggest that ERK1/2 signalling may both down-regulate members of the core ES cell regulatory network and induce transition factors [28,34–36]. Activation of ERK1/2 has also been linked to increased transcriptional output via release of paused RNA polymerase [37,38]. However, description of direct targets of ERK phosphorylation in ES cells is limited. Furthermore, control over the level and timing of ERK activity, which can profoundly influence cellular responses in other systems [39–42], has not been studied in ES cells.

Here, we applied mass spectrometry-based phosphoproteomics [43] to identify ERK-regulated proteins in ES cells. This analysis revealed a number of proteins with changes in phosphorylation status. Among these, we focused on the RSK family of protein kinases because of their potential feedback role within the ERK pathway. Genetic loss-of-function and chemical inhibition studies confirm that RSKs are major negative regulators of ERK activation. We exploit this finding to manipulate the strength of ERK signalling in ES cells and assess the consequences for entry into differentiation.

1 Wellcome Trust—Medical Research Council Stem Cell Institute, University of Cambridge, Cambridge, UK
2 Department of Biochemistry, Cambridge Centre for Proteomics, University of Cambridge, Cambridge, UK
3 Computational Proteomics Unit, Department of Biochemistry, Cambridge Centre for Proteomics, University of Cambridge, Cambridge, UK
4 Department of Biochemistry, University of Cambridge, Cambridge, UK
*Corresponding author. Tel: +44 1223 760233; E-mail: austin.smith@cscr.cam.ac.uk
†These authors contributed equally to this work
‡Present address: Therapeutics R&D Unit, Horizon Discovery, Cambridge, UK

# Results

## Global analysis of ERK-dependent phosphorylation events in ES cells

FGF4-MEK-ERK1/2 signalling stimulates ES cells to transition out of naïve pluripotency and gain competence for lineage commitment [17,44,45]. To identify targets of the ERK signalling cascade in undifferentiated ES cells, we utilised a quantitative mass spectrometry approach; stable isotope labelling by amino acids in cell culture (SILAC) [46] coupled with phosphopeptide capture using titanium dioxide ($TiO_2$) affinity purification [43]. For SILAC metabolic labelling, cells were grown in DMEM/F12 N2B27 medium supplemented with light or heavy arginine and lysine amino acids. The isotope-labelled medium was comparable in performance to standard Neurobasal/DMEM/F12 N2B27 medium [47], both in maintaining self-renewal (Fig EV1A–C) and in supporting entry into differentiation (Fig EV1D). ES cells in 2iLIF were compared to cells cultured without MEK inhibition in medium containing GSK3 inhibitor, Chir99021 (CH), plus LIF (CHLIF). This allows analysis of the ERK-responsive phosphoproteome independent of a change in identity because ES cells withstand ERK signalling and remain undifferentiated in CHLIF [7,19,20].

After withdrawal of the MEK inhibitor PD0325901 (PD) for 24 h, ES cells were sub-fractionated into two fractions by centrifugation, to increase phosphopeptide coverage; S1 comprises all organelles, the cytoplasm and the plasma membrane; N1 is enriched for nuclei (see Materials and Methods for details). Proteomes were extracted, digested with trypsin and enriched for phosphopeptides using strong cation exchange chromatography followed by $TiO_2$ affinity purification. Pooled samples were analysed on an Orbitrap Velos mass spectrometer (Fig 1A). High-throughput identification and quantitation of phosphorylated proteins from three independent experiments was performed with MaxQuant software [48]. Overall, we detected 3,248 phosphopeptide isoforms in the S1 fraction and 4,054 in N1 with a posterior error probability (PEP) of < 0.1, corresponding to 1,200 and 1,159 phosphoprotein groups, respectively, using a 1% false discovery rate (FDR). For statistical analysis of phosphorylation site changes, we selected phosphopeptides that were reproducibly identified in all three biological replicates (1,399 phosphopeptide isoforms in S1 and 2,777 in N1). Volcano plots (Figs 1B and EV1E) indicate that the majority do not show significant changes in phosphorylation site occupancy 24 h after removal of the MEK inhibitor. We detected only 22 differentially expressed phosphopeptides with consistent fold changes > 2 (adj *P*-value < 0.05), mostly in the nuclear N1 fraction. This analysis identified known ERK-interacting partners, including RSK1, Map1B [49] and Etv6 [50] that showed significant changes in phosphorylation site stoichiometry. In addition, we detected phosphorylation at Ser352 in RSK isoform 2 (*Rps6ka3*; Dataset EV1). However, this site was not significantly affected by PD withdrawal, indicating phosphorylation may be mediated by a heterologous kinase.

The complete lists of all phosphopeptides measured are presented in Dataset EV1 and EV2. The most differentially regulated phosphopeptides with adj *P*-values of 0.009, 0.017 and 0.019, respectively, were microtubule-associated protein 1B (Map1B), Jumonji domain containing 1C (Jmjd1c) and p90 ribosomal S6 kinase 1 (RSK1, *Rps6ka1*), all in fraction S1. In N1, increased phosphorylation of pluripotency-associated transcription factor Utf1 [51] was prominent.

We investigated RSK1 further because this kinase is a known regulator of ERK signalling [52–54]. It has previously been reported to be phosphorylated by ERK1/2 in ES cells [55] but a biological role has not been evaluated. RSK1 is one of four members of the mammalian RSK protein kinase family, RSK1, RSK2, RSK3 and RSK4, that are implicated in regulation of a variety of cellular processes including cell growth, proliferation, differentiation, survival and motility (reviewed in Ref. 53). RSK activation requires multiple steps of sequential phosphorylation by ERK1/2, 3-phosphoinositide-dependent protein kinase-1 (PDK1) and autophosphorylation within its functional domains. From the mass spectrometry data, we quantified an approximately fivefold increase in phosphorylation at Ser352 (Ser363 in human RSK1), an ERK target site critical for RSK activity (Fig 1C) [56]. To rule out behaviour specific to culture in SILAC medium, we examined RSK1 phosphorylation in ES cells cultured in standard medium. Immunoblotting using an antibody specific for pSer352 RSK1 confirmed the increase in pSer352 in CHLIF compared with 2iLIF. Moreover, RSK1 phosphorylation was evident at the onset of differentiation in ES cells withdrawn completely from 2iLIF for 24 h. RSK1 phosphorylation occurred without overt change in total protein expression as detected by a pan RSK1 antibody (Fig 1D).

## RSKs are major regulators of pERK1/2 levels in ES cells

Although only RSK1 phosphorylation was detected in the phosphoproteomics analysis, four RSK genes are expressed in ES cells and have potentially redundant functions [57]. Therefore, we first employed siRNA to knockdown RSK isoforms individually or in combination (Fig EV2A) and assess the consequences for pERK levels. Cells were transfected with siRNA and after 36 h transferred from 2i into N2B27. Cell lysates were prepared 1 h later for immunoblotting. Knockdown of single RSKs induced no appreciable change in levels of pERK1/2. However, depletion of RSK1 in combination with either RSK2 or RSK3 resulted in greatly elevated pERK1/2. Triple and quadruple knockdowns did not further increase pERK1/2 (Fig 2A). These results indicate that reduction in total RSK expression below a threshold level is required for substantial impact on pERK1/2. The data further suggest that RSK1 is the major effector but can be compensated by combined activity of both RSK2 and RSK3, though not by either alone.

To explore more thoroughly the requirement for RSK activity, we generated genetic mutants by multiplex genome editing using CRISPR/Cas9 [58,59]. Rex1::GFPd2 (RGd2) reporter ES cells [16] were simultaneously transfected with Cas9 and gRNAs against each of the *RSK* genes (Fig EV2B). Clones were initially screened by genomic PCR (Fig EV2C). Candidates were then expanded and examined for levels of pERK after 1 h in N2B27 (Fig 2B). We identified a number of clones that showed elevated pERK levels. Mutations were confirmed by DNA sequencing (Table EV1), immunoblotting (Fig 2C) and/or RT–qPCR (Fig EV2C). We noted that an *RSK1* null cell line (termed RSK1*23) that also carried mutations in *RSK2* and *RSK3* (but showed residual protein and mRNA expression for these genes—Fig EV2D) showed greatly increased pERK levels. Chemical inhibition of RSK activity with the small molecule BI-D1870 (BI) induced a similar elevation in pERK in parental cells. Other mutant combinations had lesser or no effects on pERK. In these mutants, addition of the RSK inhibitor BI resulted

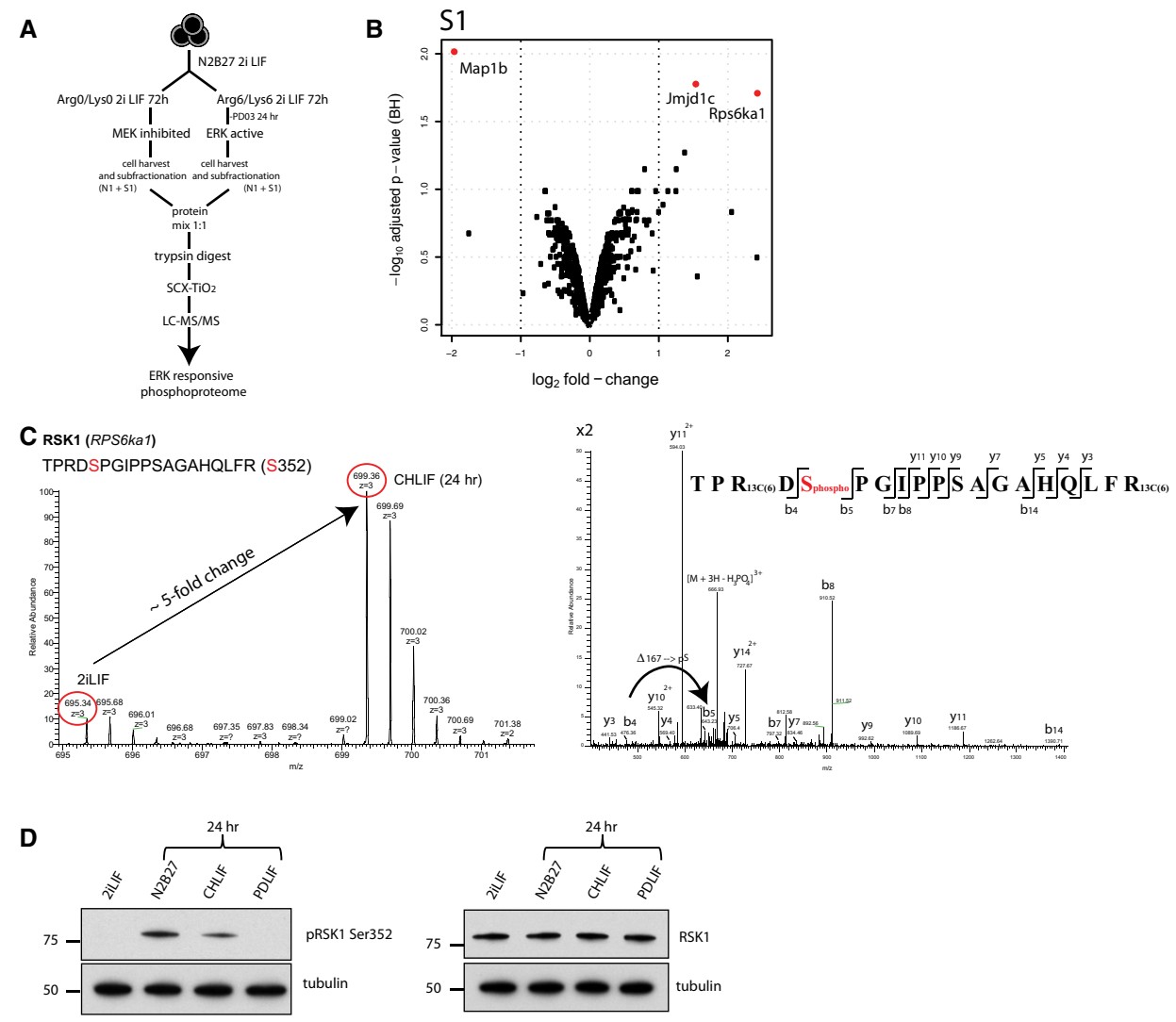

**Figure 1. The ERK-dependent phosphoproteome in ES cells.**

A Experimental workflow.

B Volcano blot illustrating fold changes and statistical significance of identified phosphopeptides from the S1 fraction. Red circles represent log2 fold changes with $P$-adj < 0.05. Results are from identifications in three independent experiments.

C MS1 and MS2 mass spectra of RSK1 phosphopeptide. Left panel: The monophosphorylated peptide exhibits an approximate fivefold increase when ERK is active. Right panel: Fragmentation mass spectrum. Phosphorylation at Ser352 could be deduced from a mass increment of 167 Da between ions $b_4$ and $b_5$.

D Immunoblots of pRSK1 (Ser352) and total RSK1 in ES cells in the indicated conditions.

Source data are available online for this figure.

in greatly increased pERK levels, consistent with persistence of substantial RSK function (Fig 2C). We performed a genetic rescue in *RSK*1*23 cells by introducing an *RSK1* transgene using piggyBac transposition (Fig EV2E). Immunoblotting showed a reduction in the pERK signal to control levels following restoration of RSK1 (Fig EV2F).

We examined the dynamics of ERK1/2 phosphorylation following 2iLIF withdrawal (Figs 2D and EV2G). In parental cells, an initial sharp increase in pERK1/2 was followed by decline within 4 h to a low level that persisted until a second major pulse of phosphorylation at 14–18 h. This pulsatile behaviour has not been observed in ES cells before but is a common feature of ERK

signalling, reflecting complex regulation [40,60]. In *RSK*1*23 ES cells, the initial peak of pERK1/2 was higher and declined more gradually and to a higher baseline. This was also followed by a second and more persistent increase from 12 to 22 h.

We then examined the consequences for expression of pERK target genes (Fig EV2H). *Egr1* expression was upregulated acutely upon 2i/LIF withdrawal and then declined by 4 h in parental cells. This on–off pattern was retained in RSK1*23 mutants, although the peak expression level was higher. In contrast, *Dusp6* and *Etv4* displayed pulsatile expression, which was more pronounced in mutants in the form of three distinct peaks. *Dusp4*, *Spry2* and *Srf* showed more cumulative gains in expression after an initial surge

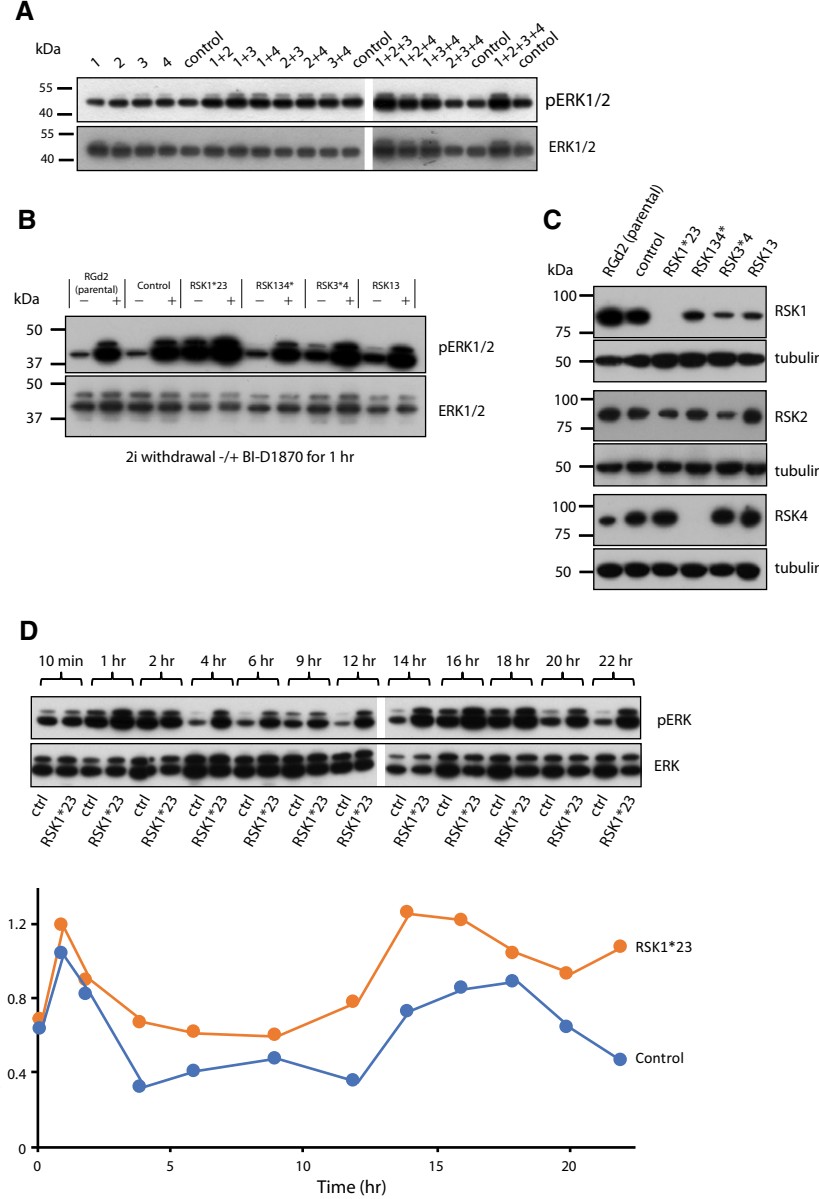

**Figure 2. RSKs are major regulators of pERK level in ES cells.**

A   Detection of pERK in RGd2 cells transfected with RSK isoform-specific siRNAs (1 = RSK1, 2 = RSK2, 3 = RSK3 and 4 = RSK4). Cells were transfected in 2i and cultured for 36 h. Medium was changed to N2B27 for 1 h before collecting lysates for immunoblotting. Controls were transfected with scrambled siRNA.

B   Immunoblot analysis of pERK in candidate clones after CRISPR/Cas9-mediated targeting. Cells were cultured in N2B27 for 1 h in the presence and absence of the RSK inhibitor BI-D1870 (BI). Nomenclature for mutants is as follows: asterisk (*) indicates null alleles; number alone indicates mutants where residual protein or mRNA is detected.

C   Immunoblot analysis of RSK1, RSK2 and RSK4 protein in mutant cell lines.

D   RSK1*23 and control (ctrl) cells were exchanged from 2iLIF into N2B27 for 22 h and cell lysates at indicated time points. Expression of pERK and ERK was detected by immunoblotting (upper panel), quantified using Fiji and the pERK/ERK ratio plotted (lower panel). Data shown are from one of two biological replicates that yielded similar results. RGd2 indicates parental line, and Control (ctrl) indicates cell line that has undergone selection process in parallel but was not edited by gRNAs.

Source data are available online for this figure.

and behaved similarly between control and mutants. Thus, individual target genes respond differently to the altered profile of pERK1/2 consequent to RSK depletion.

Collectively, these data establish that RSKs are major modulators of the ERK1/2 pathway in ES cells, although other mechanisms also contribute to the dynamic activation profile and to target gene regulation.

## RSK deficiency accelerates ES cell entry into differentiation

We examined the functional consequences of altered ERK activation in *RSK* mutant ES cells. We first monitored downregulation of GFP expression from the RGd2 reporter, which marks exit from the ES cell state [16]. Upon withdrawal from 2iLIF, loss of GFP was

more rapid in *RSK*-deficient cells than parental cells (Figs 3A and EV3A). The accelerated kinetics correlated with the observed increase in pERK levels in the order *RSK*1*23 > *RSK*13*4 > *RSK*13 > *RSK*134*. Mutant cells showed similar viability, morphology and proliferation to parental cells over the 48 h time-course (Fig 3B). However, *RSK*1*23 cells appeared slightly more elongated and flattened at the endpoint (Fig 3B), consistent with a more advanced transition. The *RSK1* transgene restored GFP downregulation kinetics, indicating that the phenotype is attributable to loss of RSK (Fig EV3C).

As a functional test of the exit from naïve pluripotency, cells cultured in N2B27 for 48 h were replated in 2i/LIF at clonal cell density for assessment of colony formation efficiency. This assay for capacity to resume self-renewal is a stringent and quantitative measure of persistent ES cell status [15]. *RSK* deleted mutant cells displayed a marked reduction in the ability to form alkaline-positive (AP) undifferentiated colonies (Fig 3C), correlating with the rate of RGd2 downregulation (Figs 2B and 3A). Reduction in colony formation was not due to intrinsic differences in plating efficiency because RSK-deficient cells maintained in 2iLIF gave rise to similar numbers of AP-positive colonies as control cells (Fig EV3B).

To test whether the effect of RSK mutation was mediated through MEK-ERK1/2 signalling, we assayed RGd2 downregulation in the presence and absence of PD (Fig 3D). Inhibition of MEK markedly impeded loss of GFP expression in *RSK*1*23 cells similar to control cells, indicating that the phenotype observed in the RSK mutants is dependent on the ERK1/2 pathway. Consistent with this, colony formation by RSK-deficient cells was restored when the cells were cultured in the presence of PD (Fig 3E).

Inhibition of GSK3 with CH is fully sufficient to sustain high GFP expression for 41 h in parental cells (Fig 3D). In RSK1*23 cells, however, GFP intensity was somewhat diminished. Moreover, colony formation after 41 h in CH was reduced twofold compared to parental cells (Fig 3E). This observation indicates that the ability of GSK3 inhibition to sustain the naïve pluripotency network can in part be overcome by increased ERK1/2 activation.

We examined whether *RSK*1*23 ES cells acquire competence to proceed further in differentiation and generate cells of different germ layers. We observed no marked differences in the ability of *RSK*1*23 ES cells to differentiate towards Sox1-positive neural progenitors and Tuj1-positive neurons under autocrine stimulation (Fig 3F). Furthermore, RSK1*23 mutants could respond to inductive regimes and generate either T/Eomes double-positive mesoderm progenitors (Fig 3G) or Foxa2/Sox17 double-positive definitive endoderm cells with high efficiency (Fig 3H).

We conclude that abrogation of RSK activity accelerates but does not compromise the ES cell exit from naïve pluripotency and transition to multilineage competence.

## RSK deficiency accelerates dissolution of the naïve pluripotency circuitry

Progression through pluripotency is marked by sequential downregulation of naïve factors and upregulation of peri-implantation epiblast markers, followed by appearance of lineage specification markers [16,17,61,62]. We examined levels of naïve pluripotency

transcription factors in *RSK*1*23 cells during transition. We assessed protein expression for Esrrβ, Klf2, Klf4, Tfcp2l1 and Nanog by immunoblotting between 40 and 48 h after withdrawal from 2iLIF (Fig 4A). We observed earlier downregulation of each of these transcription factors in *RSK*1*23 cells compared to parental cells. Tfcp2l1 protein was still detectable in parental cells up to 42 h but was absent from the mutants by 40 h. Klf2, Klf4 and Nanog were present in parental cells at 48 h, but substantially reduced in mutants at 40 h and undetectable from 44 h. Esrrβ persisted in mutants, but at much lower levels than in parental cells. These data show that the naïve transcription factor circuitry collapses more rapidly in response to ERK1/2 activation in the *RSK*1*23 mutants. In contrast, the core pluripotency transcription factor Oct4 remained largely unaffected by higher levels of pERK and was still strongly expressed at 48 h, consistent with its maintained expression throughout pluripotency progression.

We then examined markers of developmental progression by RT–qPCR. We tested expression of two transitional pluripotency markers *Oct6* and *Fgf5* that are upregulated throughout the peri-implantation epiblast [61,63,64]. We observed that *RSK*1*23 cells showed earlier upregulation of both of these markers (Fig 4B). ERK1/2 signalling also plays a role in extraembryonic lineage specification in the early embryo, and we therefore examined two markers for this lineage, *Gata6* and *Gata4* [65–67]. Expression remained negligible (Fig 4B), confirming that in these defined conditions, extraembryonic endoderm is not induced by the ERK1/2 pathway [16].

These findings suggest that restriction of ERK activity by RSK proteins constrains the rate of ES cell transition into an early post-implantation epiblast-like state.

## Pharmacological inhibition of RSK increases pERK1/2 and expedites ES cell lineage specification

The small molecule kinase inhibitor BI-D1870 (BI) has selectivity for RSK and has been found to increase ERK1/2 phosphorylation in rat embryonic fibroblasts [68]. We investigated the effect of 3 μM BI on ES cells after withdrawal from 2i. ERK activation was monitored by immunoblotting (Fig 5A). We observed substantially increased phosphorylation of ERK1/2 in the BI-treated cultures, with a peak at 30 min followed by gradual decline. The mobility shift in the BI-treated samples indicated that the vast majority of total ERK1/2 proteins were phosphorylated at early time points. We also noted modestly reduced GSK3 phosphorylation in BI-treated cells (Fig 5A). This is consistent with the known action of RSK to phosphorylate the inhibitory N terminus of GSK3α/β at Ser9 and 11, thereby creating a pseudosubstrate domain that can bind to and block the active site [69].

We analysed two well-characterised early immediate response genes to ERK signalling, *Egr1* (early growth response 1) and *Dusp6* (dual specificity phosphatase 6). Both genes showed a significant upregulation 3 h after BI treatment (Fig 5B).

We investigated whether BI treatment had consequences for ES cell transition kinetics in line with the *RSK*1*23 mutant phenotype. We applied 3 μM BI to RGd2 cells for 29 h after withdrawal from 2i and monitored the time-course of GFP downregulation. BI-treated cells shifted to the GFP low state more rapidly and uniformly

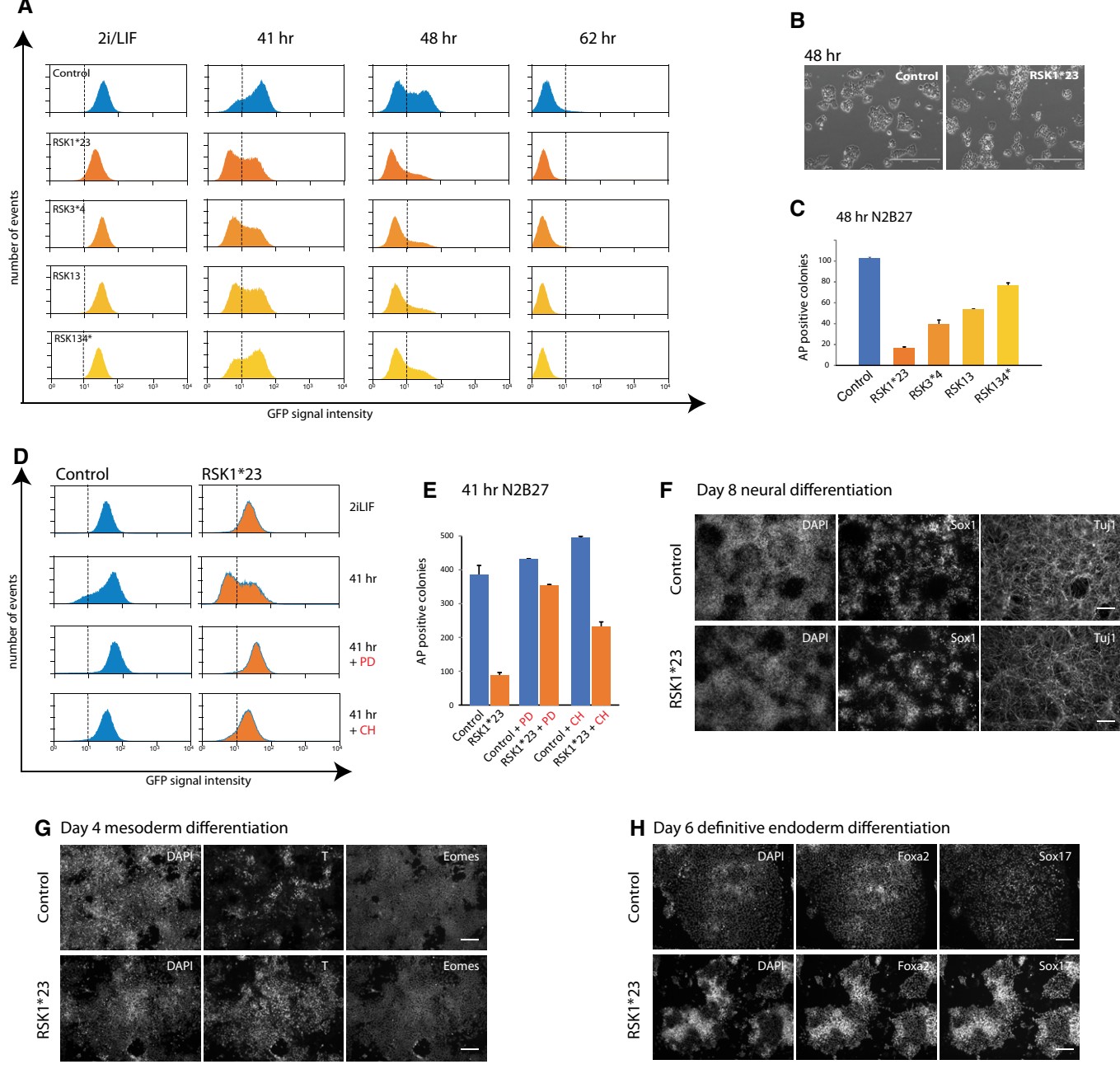

**Figure 3.  Deletion of *RSK1* accelerates ES cell transition.**

A   Flow cytometry analysis of GFP downregulation in RGd2 parental cells and indicated RSK mutants following withdrawal of 2iLIF.

B   Phase contrast images of RGd2 parental and RSK1*23 cells cultured in N2B27 for 48 h. Scale bar: 400 μm.

C   Colony-forming assay on *RSK* mutant cell lines after withdrawal from 2iLIF for 48 h. Six hundred dissociated cells were plated per 6 wells in 2iLIF. Plot shows numbers of undifferentiated alkaline phosphatase (AP)-positive colonies stained after 5 days. Mean and SD shown; *n* = 2.

D   Flow cytometry analysis of RGd2 expression in parental line and RSK1*23 mutant cells in the presence of PD0325901 (PD) or Chir99021 (CH) for 41 h.

E   Colony-forming assay on *RSK*1*23 cells cultured in PD or CH for 41 h. One thousand cells were plated per 6 wells in 2iLIF and stained for AP after 5 days. Mean and SD shown. *n* = 2.

F   Immunostaining of *RSK*1*23 cells with Sox1 and Tuj1 antibodies after 8 days culture in N2B27. Scale bar: 100 μm.

G   Immunostaining of RSK1*23 cells with T and Eomes after 4 days in N2B27 medium supplemented with ActivinA and CH. Scale bar: 100 μm.

H   Immunostaining of RSK1*23 cells with Foxa2 and Sox17 antibodies after 6 days in definitive endoderm inducing media. Scale bar: 100 μm.

(Figs 5C and D, and EV4A). Moreover, after 25 h of BI treatment of cells, colony formation was reduced upon replating in 2iLIF (Fig 5E), corroborating accelerated transition.

Interestingly, addition of BI during 2i withdrawal resulted in faster downregulation of Nanog protein and mRNA (Fig EV4B and C). Klf4 protein and mRNA were also downregulated faster in BI,

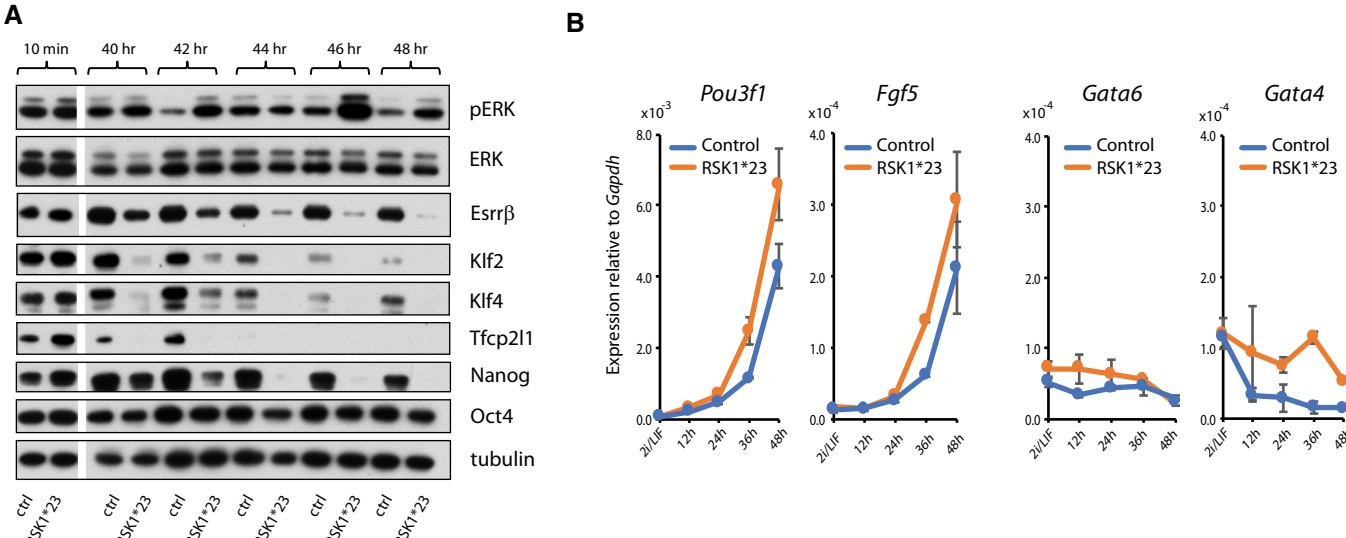

**Figure 4.  RSK mutations increase the rate of entry into differentiation.**

A   RSK1*23 and parental cells were withdrawn from 2iLIF and cell lysates prepared between 40 and 48 h. Immunoblotting was performed with indicated antibodies. Ctrl = parental line.

B   Marker gene expression analysis by RT–qPCR in *RSK1*23* mutants and parental cells transferred from 2iLIF into N2B27 for indicated times. Mean and SD shown; *n* = 2.

Source data are available online for this figure.

although the difference was more modest. The downregulation of Tfcp2l1 and Klf2 mRNA did not appear to be accelerated by BI.

We then examined the time window over which transition is sensitive to RSK inhibition. We found that BI treatment for 6–25 h, 12–25 h or 18–25 h resulted in a similar outcome to 25 h continuous treatment (Fig EV4D). However, treatment during the first 6 or 12 h only had negligible effects (Fig EV4D). The effectiveness of RSK inhibition during the 18- to 25-h interval suggests that the modulation of ERK activity close to or during transition is key to the efficiency of exit from naïve pluripotency.

We evaluated whether BI treatment affected subsequent neural differentiation, which predominates in adherent culture in the absence of exogenous inducers [47]. Cells were transferred from 2i to N2B27, and BI was applied for 25 h. We analysed cells on days 2 and 3, when Oct4-negative/Sox1-positive neural precursors begin to emerge [17,33,70]. No significant effects were observed on day 2. On day 3, however, cells treated with 3 μM BI for 25 h showed a striking increase in the proportion of Sox1-positive cells and a much lower proportion of Oct4-positive cells by immunostaining (Fig 5F and G). Application of BI at different time intervals had a less significant effect (Fig EV4E).

To confirm the neural differentiation potential of cells generated after BI treatment, cultures were maintained in N2B27 for 7 days and then stained with TuJ1 to detect immature neurons. We observed that the majority of cells became TuJ1 immunoreactive and many displayed extended processes typical of post-mitotic neurons (Fig 5H).

Finally, we investigated whether BI treatment would affect the multilineage differentiation capacity of cells. Cells were plated in ActivinA and CH after 25 h of N2B27 + BI treatment to initiate primitive streak-like differentiation [71–74]. On day 3, BI-treated cells upregulated Brachyury similar to untreated controls (Fig 5I). We

subsequently applied conditions promoting definitive endoderm differentiation [72] and stained for Foxa2. BI-treated cells showed robust expression of this marker in the majority of cells on day 5 (Fig 5J). Therefore, cells treated with the RSK inhibitor are not restricted to neural differentiation but display multilineage potency.

Overall, these data indicate that short-term RSK inhibition with the small molecule BI can expedite developmental progression from the naïve ES cell state.

## Discussion

Genetic loss-of-function and pharmacological inhibition studies have previously established a role for ERK1/2 signalling as a driver of mouse ES cell transition towards lineage competence. Whether ERK1/2 signal amplitude, duration or periodicity are important parameters has been unexplored, however. Active ERK1/2 has been challenging to manipulate because the pathway is stimulated in an autocrine manner by FGF4 and buffered by multiple feedback mechanisms [75–78]. Our study identifies RSK proteins, and in particular RSK1, as the major feedback regulators that modulate pERK1/2 levels in ES cells.

RSK function in negative feedback regulation of the ERK pathway is well documented. In *D. melanogaster,* RSK regulates eye and wing vein development, which is controlled by the Ras-MEK-ERK pathway [76]. Genetic manipulations in *Xenopus laevis* have suggested that RSK4 negatively regulates mesoderm formation by inhibiting the RAS-MEK-ERK pathway downstream of receptor tyrosine kinases [77]. That report further described an inverse correlation between RSK4 and active ERK1/2 expression in extraembryonic tissue of the mouse embryo. Increased pERK was also found in skeletal tissue of RSK2-deficient mice [75].

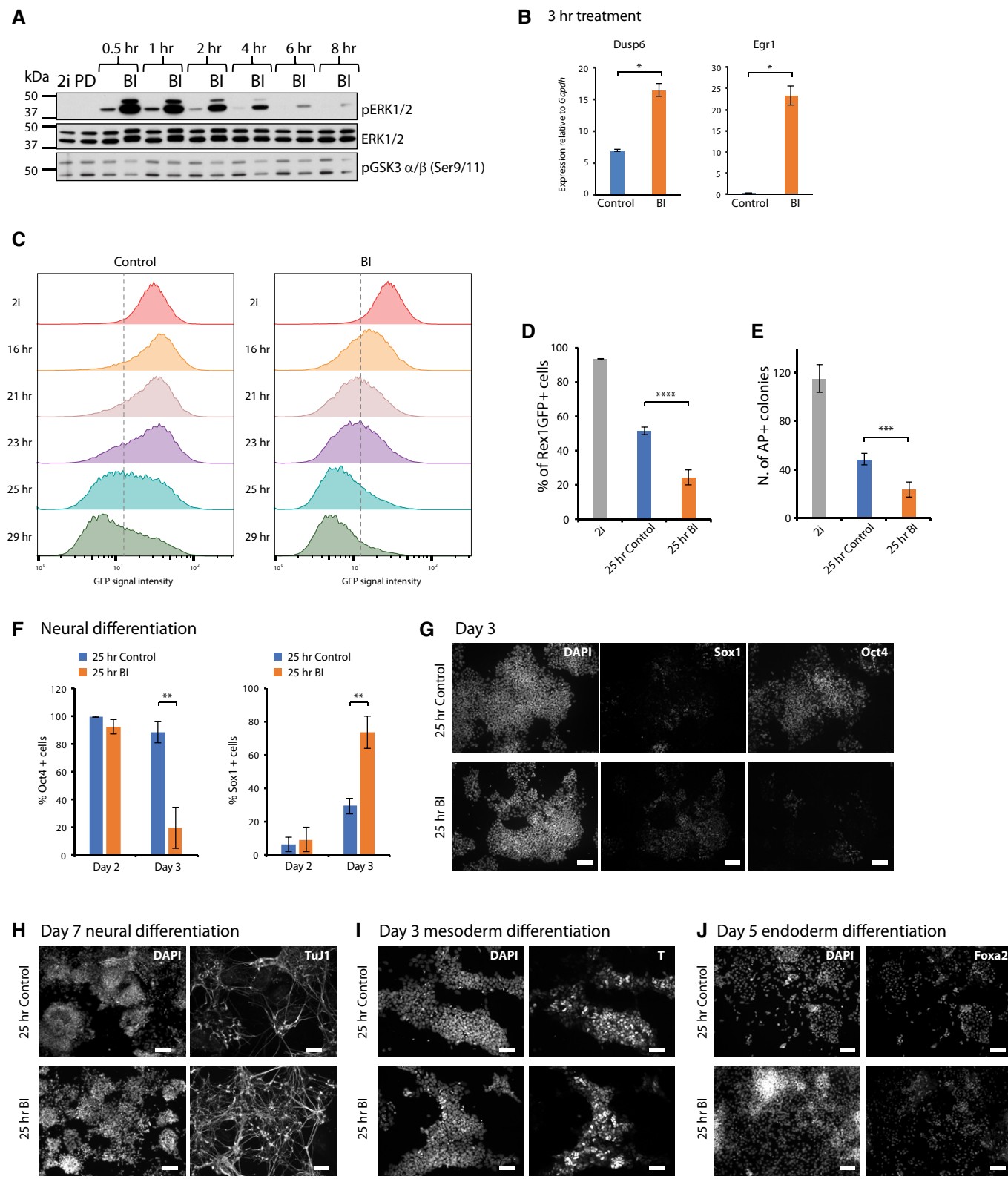

**Figure 5.**

Biochemically, reduction in pERK by RSKs is thought to be mediated through phosphorylation of SOS [52]. A possible mechanism has been suggested whereby phosphorylated SOS1 binds to 14-3-3 proteins, resulting in reduced activity of SOS1 or prevention of its interaction with the signalling complex [54]. RSK has also been shown to restrict ERK activation by phosphorylating the

◄

**Figure 5.  Small molecule inhibition of RSKs increases pERK and accelerates differentiation.**

A     RGd2 cells were cultured with and without 3 μM BI after withdrawal from 2i. Cell lysates were obtained at indicated time points and immunoblotted.
B     Expression analysis of immediate early genes in BI-treated RGd2 cells at 3 h. Mean and SD shown; $n$ = 2; unpaired two-tailed Student's $t$-test.
C     Flow cytometry time-course on RGd2 cells withdrawn from 2i and treated with BI.
D     Proportion of RGd2-positive cells at 25 h when BI was added at different intervals. Mean and SD shown; $n$ = 3; unpaired two-tailed Student's $t$-test ****$P < 10^{-5}$.
E     Colony-forming assay after 25 h of BI treatment. Mean and SD shown; $n$ = 3; unpaired two-tailed Student's $t$-test ***$P < 10^{-4}$.
F     Quantification of Oct4 and Sox1 immunostaining at day 2 and day 3 of neural differentiation following BI treatment. Mean and SD shown; $n$ = 3; unpaired two-tailed Student's $t$-test **$P < 5 \times 10^{-3}$.
G     Representative images of day 3 neural differentiation without or with BI treatment. Scale bar: 100 μm.
H–J   Immunostaining for neural (TuJ1), mesendoderm (T) and endoderm (Foxa2) lineage markers following BI treatment. ES cells were treated or not with BI for 25 h and then subjected to lineage specification protocols. Staining was performed on day 7 with TuJ1 (H, scale bar: 100 μm), on day 3 for T (I, scale bar: 50 μm) and on day 5 for Foxa2 (J, scale bar: 100 μm).

Source data are available online for this figure.

Grb2-associated binder Gab2, preventing SHP2 recruitment and coupling to the MEK-ERK pathway [78].

Both the peak levels and the baseline activity of ERK1/2 are increased in RSK-deficient and BI-treated ES cells. The persistence of peaks indicates that other feedback mechanisms, such as the dual specificity kinases [28], are also operative. ERK1/2 signalling is not involved in cell cycle progression in mouse ES cells [79]. Indeed, the ERK pathway is not required for ES cell propagation [6] and its activation is associated with entry into differentiation [27,33]. While some differentiation events can occur without ERK1/2 signalling [20,31], the finding that RSK deletion or inhibition accelerates ES cell transition adds to the weight of evidence that ERK1/2 signalling acts as a trigger for exit from self-renewal. Nonetheless, RSK has additional targets that may also play a role. Notably, it can phosphorylate GSK3 leading to a decrease in activity [80]. Inactivation of RSK could therefore lead to decreased intracellular β-catenin and increased repression of Esrrb and other naïve factors by Tcf3 [19]. However, the observations that RSK mutant ES cells differentiate even in the presence of GSK3 inhibitor and that Esrrb downregulation is not markedly accelerated indicate that the major effect is through pErk.

Reducing RSK activity with the small molecule inhibitor BI enhanced activation of pERK1/2 resulting in earlier clearance of the naïve pluripotency gene regulatory network and subsequently more rapid lineage specification. Notably, BI treatment accelerates loss of ES cell identity and function by up to 6 h without compromising differentiation potential. In adherent culture in the absence of serum factors, ES cells progress from naïve pluripotency into a population that resembles the early post-implantation formative epiblast [16,45,81,82] with competence for germline and somatic lineage specification [17,45,81,83]. This process is asynchronous, with individual ES cells departing from the naïve state at different times [16,84,85]. The findings here indicate that pERK1/2 activity is limiting for progression from the naïve state. The facility to accelerate the rate of ES cell differentiation by short-circuiting negative feedback with a small molecule RSK inhibitor is potentially a valuable tool for reducing the time window of transition (Fig 5C and D) and thereby facilitating analyses of pluripotency progression.

It is noteworthy that the second peak of pERK1/2 after 2i withdrawal occurs shortly before the first ES cells to exit the naïve state are robustly detected at 20 h [44] and that BI administered during this time alone is sufficient to increase the frequency of exit. These observations may suggest that the second pulse of pERK1/2 is a critical event. It will be of interest to determine specific downstream consequences at this time.

In summary, our results establish that modulation of the ERK activation profile has a quantitative effect on the dynamics of transition from naïve pluripotency. Reducing negative feedback to enforce more homogenous exit from self-renewal could potentially contribute to improving the overall efficiency of directed lineage commitment. In contrast, in the context of the embryo, feedback modulation of ERK activity by RSK family members and other regulators may provide flexibility in the rate of developmental progression and safeguard against premature depletion of the naïve founder compartment.

# Materials and Methods

## ES cell culture

Rex1::GFPd2 (RGd2) ES cells carry a destabilised form of GFP with a short half-life under the control of the endogenous Rex1 (*Zfp42*) promoter. *Zfp42* expression is associated with the naïve state and quickly downregulated as cells lose the capacity to self-renew in 2iLIF conditions [44]. Cells were cultured in the absence of feeders on 0.1% gelatin-coated plastic and replated at a density of $2 \times 10^4$ cells/cm² every 2–3 days following dissociation with accutase (PAA, cat. L11-007) or trypsin (Invitrogen, cat. 25050030). Cells were grown in either GMEM medium (Sigma, cat. G5154) containing 15% FCS (Sigma, cat. F7524), 100 mM 2-mercaptoethanol (Sigma, cat. M7522), 1× MEM non-essential amino acids (Invitrogen, cat. 1140-036), 2 mM L-glutamine, 1 mM sodium pyruvate (both from Invitrogen) and 100 units/ml LIF prepared in-house, or in serum-free N2B27 [86] (prepared in-house, or NDiff N2B27 base medium, Stem Cell Sciences Ltd, cat. SCS-SF-NB-02, or NDiff 227 base medium, Takara Bio, cat. Y40002) supplemented with small molecule inhibitors PD (1 μM, PD0325901), CH (3 μM, CHIR99021) and LIF. For indicated experiments, cells were treated with 3 μM BI-D1870 inhibitor (BI; Axon Medchem). For metabolic labelling, cells were cultured in arginine- and lysine-free DMEM/F12 (Dundee Cell Products) supplemented with B27 (Gibco cat. 17504-044), in-house prepared N2 [86], 100 mM 2-mercaptoethanol, 2 mM L-glutamine, 0.7 mM L-arginine (Sigma) or $^{13}$C$_6$ L-arginine (CK Gas Products), 0.5 mM L-lysine (Sigma) or $^{13}$C$_6$ L-lysine (CK Gas Products) and the two inhibitors PD and CH (2i) plus LIF. For colony formation assays, cells were plated on laminin-coated plates in 2iLIF for 5 days and fixed and stained for alkaline phosphatase (Sigma, cat. 86 R-1KT) according to the manufacturer's instructions. Plates were

scanned using a CellCelector (Aviso) instrument, and colonies were scored manually using ImageJ software.

## Phosphoproteomics analysis

ES cells were labelled in light (Arg0/Lys0) and heavy (Arg6/Lys6) SILAC medium for 72 h before removing PD from the heavy cell culture for 24 h. Cells were collected by dissociation with Accutase containing 2iLIF for the light culture and CHLIF for the heavy culture and cell numbers determined using a haemocytometer. Cells were resuspended in ice-cold fractionation buffer [0.25 M sucrose, 50 mM Tris–HCl, pH 7.9, 5 mM EDTA, 10 mM DTT, PhosSTOP Phosphatase Inhibitor Tablet (Roche), EDTA-free Protease Inhibitor Tablet (Roche)] containing 2iLIF for the light cells and CHLIF for the heavy cells at a density of $1 \times 10^7$ cells/ml. Typically, $0.5–1 \times 10^8$ cells were used per experiment. Absence of cell lysis was checked using phase contrast microscopy before transfer into a pre-chilled cell disruption bomb (Parr, model 4639) [87]. Cell suspensions were incubated at 175 psi for 10 min on ice and then adiabatically decompressed via drop wise release from the vessel. Cell disruption was assessed by microscope and showed that almost all nuclei (95–100%) were released. Nuclei-enriched fractions (named N1) were obtained by centrifugation at 600 g for 10 min and snap-frozen in liquid nitrogen before storage at −80°C. The remaining cell material consisting of all other organelles, the cytoplasm and the plasma membrane (termed S1 fraction) was incubated with RIPA lysis buffer for 10 min on ice before spinning at 2,800 g at 4°C for 10 min to pellet cell debris. The supernatant of the S1 fraction was transferred to a new tube, and proteins were precipitated with 4× volumes of ice-cold 80% acetone at −20°C overnight. N1 samples were thawed on ice and membranes disrupted by addition of 2× RIPA lysis buffer in volumes of 0.5–1 ml. Lysed N1 fractions were sonicated and proteins precipitated as described before. Protein pellets of S1 and N1 fractions were washed with ice-cold water by vortexing rigorously followed by centrifugation for 30 min at either 4,000 g for S1 or at maximum speed of a benchtop centrifuge for N1. The washing step was repeated once more, and proteins from both fractions were resolubilised in 200 µl of 8 M Urea containing 500 mM TEAB. Protein concentrations were determined using Pierce BCA protein assay in a 96-well plate format according to the manufacturers' instructions. At that point, light and heavy N1 samples were mixed at a 1:1 ratio as well as light and heavy S1 fractions at the same ratio, which generally yielded around 3 mg of protein per sample.

Proteins were reduced with DTT (20 mM final) for 35 min at room temperature followed by alkylation with IAA (40 mM final) for another 35 min at room temperature in the dark. Samples were diluted 1:10 with water, and trypsin (Worthington) was added at an enzyme/substrate ratio of 1:50. Trypsin digestion was performed on a shaking platform for 5 h at 37°C in a hybridisation oven before snap-freezing on dry ice/ethanol and lyophilisation for 3 days. Dried peptides were resuspended in 0.5–1 ml of solvent A (5 mM $KH_2PO_4$/ 30% acetonitrile, pH 2.7) for SCX separation [88] on a fully automated nanoAcquity UPLC system operated with MassLynx MS software (Waters). Peptides were repeatedly injected onto a 4.6 mm × 20 cm column (Poly LC, Columbia, MD) containing 5-µm polysulfoethyl aspartamide beads with a 200-Å pore size using a 250-µl injection loop (Waters). Separation was performed with a gradient consisting of 2 min at 100% solvent A, 15 min gradient to 15%

solvent B (solvent A with 350 mM KCl), 20 min gradient to 100% solvent B and 15 min at 100% solvent B. The column was then washed and recalibrated using a gradient of 15 min at 100% C (0.1 M Tris/0.5 M KCl, pH 7.0) followed by 10 min at 100% D (water) and 20 min of 100% A. UV was monitored with a PDA detector (Waters) operating within a range of 190 and 500 nm, and the flow rate was set to 0.5 ml/min. Sample collection was every 2 min, and the first 20 fractions were lyophilised and desalted as described in [89]. Salt-free samples were then stored at −80°C until phosphopeptide enrichment.

Each SCX fraction was redissolved in 60 µl of 100 mg/ml 2,5 dihydroxybenzoic acid dissolved in 80% acetonitrile and 6% trifluoroacetic acid (loading buffer) and incubated for a minimum of 5 and a maximum of 15 min with ∼ 2 mg of $TiO_2$ beads that had been pre-washed in the same solution. The beads were then washed with 60 µl of loading buffer followed by two washes with 1 ml of 50% acetonitrile and 0.1% trifluoroacetic acid, each washing step shaking for 5 min. Phosphopeptides were eluted by adding 50 µl of ammonia water (pH ≥ 10.5) directly to the beads and incubating on the shaker for 5 min. The eluate was then carefully removed into a fresh tube containing 50 µl of 20% formic acid and stored at −20°C until identification by mass spectrometry. For LC-MS/MS analysis, samples were dried down in a Speed Vac until approximately 10 µl of the solution remained and 5 µl was transferred into an autosampler vial and diluted 1:2 with 20% formic acid. About 8 µl of this peptide mixture was analysed by the mass spectrometer.

Peptide solutions were injected onto a 180 µm × 20 mm (5 µm particle size) C18 trap column (Waters UPLC Trap Symmetry) from the nanoAcquity sample manager of a fully automated nanoAcquity UPLC system (Waters) using 0.1% formic acid in water (Buffer A) at a flow rate of 10 µl/min. Peptides were then separated on a 75 µm × 250 mm (1.7 µm particle size) reverse-phase BEH C18 analytical nano-column (Waters) at a flow rate of 300 nl/min using a binary gradient consisting of buffer A and buffer B (0.1% formic acid in acetonitrile). The HPLC system was directly coupled to a LTQ Orbitrap Velos instrument (Thermo Scientific) operating at a resolving power of 60,000 and equipped with a New Objective nanospray ionisation source. Peptides were eluted from the column with a linear gradient from 5 to 45% buffer B over 45 min, followed by a wash step and re-equilibration step, giving a total running time of 60 min. In the Orbitrap analyser, a survey scan was performed over a mass range of m/z 380–1,500 each of them triggering 5 MS2 LTQ acquisitions of the 5 most intense ions exceeding 500 counts using a data-dependent acquisition mode. Peptide ions with charge states of $2^+$ and above were automatically isolated for MS/MS in the LTQ linear ion trap and fragmented by collision-induced dissociation. The Velos mass analyser was internally calibrated on the fly using the lock mass of polydimethylcyclosiloxane at m/z 445.120025.

Raw data files obtained from the LTQ Orbitrap Velos were loaded into MaxQuant software version 1.3.0.3 using the programs' standard settings for protein identification and quantitation, except for the following: "maximum charge" state for proteins was set to 4; "MS/MS tolerance" for FTMS was set to 0.8 Da; "minimum razor + unique peptides" was set to 0; "minimum unique peptides" was set to 1 for protein identification; "filter labelled amino acids" was deselected; and "use unique peptides" was selected for protein. The data files were searched against the Uniprot mouse database (January 2012; 85,688 sequences) using Andromeda software within the MaxQuant software package. Phosphopeptides with a PEP < 0.1 that were

quantified in the three replicates (1,399 and 2,777 phosphopeptides for S1 and N1, respectively) were analysed to identify differences in expression using linear models and empirical Bayesian model as implement in the Bioconductor [90] package limma [91,92], and *P*-values were adjusted using the Benjamini and Hochberg method [93].

## Immunoblotting

Typically, ~ $1 \times 10^6$ cells were lysed with ice-cold RIPA lysis buffer [10 mM Tris–HCl, pH 7.9, 30 mM NaCl, 5 mM EDTA, 0.2% NP-40, 0.2% sodium deoxycholate, 0.2% sodium dodecylsulfate [SDS], EDTA-free protease inhibitor tablet and PhosSTOP Phosphatase inhibitor tablet (Roche)], sonicated for $3 \times 30$ s on ice with 30-sec intervals with an ultrasonic liquid processor (Misonix) and protein concentration was determined using Pierce BCA protein assay in a 96-well plate format according to the manufacturers' instructions. Plates were read with SoftMax Pro software using a Spectramax M2e plate reader (Molecular Devices). Approximately 5 μg of protein was mixed with the appropriate volume of 5× Laemmli sample buffer and DTT (10 mM final), heated and separated on a precast Novex 4-12% Bis–Tris SDS–polyacrylamide electrophoresis gel. After electrophoresis, proteins were transferred onto a PVDF membrane and blocked with 4% bovine serum albumin in Tris-buffered saline (TBS; 150 mM NaCl, 50 mM Tris–HCl, pH 7.9) containing Triton X-100 (0.2%) for 1 h at room temperature. Primary antibodies were diluted in blocking solution and incubated with the membrane overnight at 4°C. After washing the membrane, species-specific and HRP-conjugated secondary antibodies (Amersham) were used for detection with ECL plus Western blotting detection reagent (Amersham) on X-ray film (Kodak) according to the manufacturer's instructions. ImageJ was used to quantify the pERK1/2 vs. ERK Western blots. Briefly, boxes of equal size were drawn around each band and the mean intensity calculated. The ratio of pERK/ERK intensity over time was plotted. Primary antibodies were RSK1 (Cell Signalling Technology, 9333) 1:1,000; Phospho-RSK1 (*pSer352*; Sigma, SAB4300092) 1:500; Phospho-p44/42 MAPK (Erk1/2; Thr202/Tyr204; Cell Signalling Technology, 4370) 1:500; p44/42 MAPK (Erk1/2; Cell Signalling Technology, 9102) 1:1,000; RSK2 (Cell Signalling Technology, 9340) 1:1,000; RSK4 (Santa Cruz, sc-100424) 1:200; Esrrβ (Perseus, PP-H6705-00) 1:1,000; Klf2 (kind gift from Huck-Hui Ng laboratory) 1:3,000; Klf4 (R&D, AF3158) 1:500; Tfcp2l1 (R&D, AF5726) 1:1,000; Nanog (Cosmobio, REC-RCAB002P-F) 1:2,000; Oct3/4 (C10, Santa Cruz, sc-5279) 1:1,000; Phospho-GSK-3α/β (Ser21/9; Cell Signalling Technology, 9331) 1:1,000; and alpha-tubulin (Abcam, ab4074) 1:30,000.

## Gene expression analysis by RT–quantitative PCR

Total RNA was prepared with the RNeasy Kit (Qiagen), and 500 ng was reverse-transcribed using SuperScriptIII (Invitrogen) and oligo-dT primers according to the manufacturer's protocol. Real-time PCR was performed using TaqMan Fast Universal Master Mix and TaqMan probes (Applied Biosystems) or the Universal Probe Library (UPL, Roche) system. *Gapdh* or *Actb* were used as an endogenous control (Applied Biosystems). For siRNA and rescue experiments, the data were further normalised to control cell line. All experiments were performed in biological duplicate if not otherwise indicated. Results are shown as mean and standard deviation. TaqMan probes were *Pou5f1* (Oct4), Mm00658129_gH; *Klf4*, Mm00516104_m1; *Klf2*,

Mm01244979_g1; *Nanog*, Mm02384862_g1; *Rex1*, Mm03053975_g1; *Pou3f1* (Oct6), Mm00843534_s1; and *Fgf5*, Mm03053745_s1. Primer sequences and UPL probe numbers are listed in Table EV2.

## Immunostaining

Cells were fixed in chilled 4% PFA and washed with PBS. Cells were blocked with 3% donkey serum in PBS-Tween (0.1%) and incubated with primary antibody overnight at 4°C. Cells were washed with PBS-Tween and incubated with secondary antibody in blocking solution for 2–3 h in the dark at room temperature. Cells were washed and nuclei-counterstained with 5 μg/ml DAPI. Immunofluorescence and phase contrast images were acquired on a Leica DMI 4000B microscope and processed using Las AF software. Primary antibodies were Oct4 (Santa Cruz, sc-5279, sc-8628) 1:200; βIII-tubulin (Tuj1; Covance, MMS-435P) 1:500, Pax6 (Developmental Studies Hybridoma Bank) 1:100; Nanog (eBiocsciences, 14-5761) 1:200; Klf4 (R&D AF3158) 1:400; Brachyury (T; R&D AF2085) 1:300; Foxa2 (Abcam, ab23630) 1:200; Sox17 (R&D, AF1924); Eomes (Abcam, ab23345); and Sox1 (Cell Signalling, 4194) 1:200. Species-specific or Ig-subtype-specific Alexa Fluor 488-, 555- and 647-conjugated secondary antibodies (1:1,000; Invitrogen) were used. To quantify immunostainings, multiple fields were taken at random for each condition/day and analysed by CellProfiler [94]. The DAPI channel was used to identify nuclei as regions of interest to determine fluorescence level for the remaining channels. The threshold fluorescence for positive/negative staining was determined either by staining samples only with secondary antibodies or by staining-negative cells with both primary and secondary antibodies.

## Flow cytometry

Flow cytometry analyses were performed using a CyAn ADP flow cytometer (Dako) and processed with Summit software (V4.3.02). For monitoring, Rex1::GFPd2 (RGd2) expression in the presence and absence of inhibitors cells was plated at $1.5 \times 10^4$ cells/cm$^2$ in 2i for 24 h before inhibitor withdrawal and addition of BI at the times indicated. For measuring RGd2 in the RSK mutant cell lines, cells were plated at $2 \times 10^4$ cells/cm$^2$ on 0.1% gelatin-coated plastic in 2iLIF for 24 h before withdrawal of the inhibitors and LIF as indicated. Viable cells were gated on FSC-A vs. SSC-A.

## siRNA transfection

Single and combinatorial transfections were performed in 12-well plates (4 cm$^2$) using 1 μl of Dharmafect 1 (Dharmacon, cat. T-2001-01) reagent, 1 μl of siRNA solution (20 μM) for each gene and $2 \times 10^5$ cells in 1 ml of 2i medium. The medium was replaced after overnight incubation with fresh 2i and cultured for another 24 h before transfer into N2B27 for 1 h. siRNAs were purchased from Qiagen (Flexitube GeneSolution, see Table EV3).

## DNA transfection

For RSK1 overexpression, the full length coding sequence of RSK1 was amplified by PCR from ESC cDNA and cloned into a gateway Donor vector using pENTR™D-TOPO® Cloning Kit (Invitrogen). The cDNA clone was confirmed by sequencing and then cloned into

a PiggyBac destination vector pPB-CAG-DEST-PGK-Hygromycin to make the pPB-CAG-RSK1-PGK-Hygromycin expressing vector driven by a CAGGS promoter. To generate RSK1-overexpressing ES cells, 0.3 μg of pPB-CAG-RSK1-PGK-Hygromycin and 0.3 μg of PBase were co-transfected into $3 \times 10^5$ cells using 3 μl Lipofectamine 2000 (Life Technologies). Transfections were carried out in FCS-containing medium supplemented with 2iLIF. 200 μg/ml of hygromycin was applied for more than 5 days to select for stable integration of pPB-CAG-RSK1-PGK-Hygromycin.

### CRISPR/Cas9 targeting

Guide RNA (gRNA) sequences were designed to target the third exons of *Rps6ka1*, *Rps6ka*2, *Rps6ka*3 and *Rps6ka*6 using the online resources provided at http://tools.genome-engineering.org (see Fig EV2B and Table EV2). gRNAs were cloned into a U6 expression vector (kind gift from Sebastian Gerery) and transfected into $3 \times 10^5$ RGd2 cells together with 0.3 μg Cas9 nuclease and 0.1 μg dsRed containing plasmids using 3 μl Lipofectamine 2000. After 48 h, single dsRed-positive cells were deposited by FACS into 96-well plates using a MoFlo instrument (Dako). Surviving colonies were expanded in FCS-containing medium supplemented with 2iLIF and clones screened by genomic PCR for CRISPR/Cas9-induced mutations in the targeted locus. Two primers were designed to amplify genomic regions spanning the gRNA target sites (highlighted in red in the Table EV2). A third primer was designed to align with its 3′ end to the third nucleotide upstream of the PAM site. Triple primer PCR discriminated between mutant and wild-type alleles (Fig EV2C). Potential homozygous clones were further expanded for analysis of protein expression. The PCR amplicons from primers that span the gRNA target site (highlighted in red in the Table EV2) were cloned into pCR-Blunt II-TOPO vectors (Invitrogen). Four to eight colonies were picked and sequenced by Sanger sequencing. The results are summarised in Table EV1. Table EV1 shows a summary of the status of each clone used in this study.

### *In vitro* differentiation

For neural and mesoderm differentiation, cells were plated in 2i at $1 \times 10^4$ cells/cm$^2$ onto laminin- or fibronectin-coated plates, respectively. After 24 h, cells were washed once with PBS before changing the media to N2B27 with or without BI for the indicated period. After 24 h, media were changed to fresh N2B27 for neural differentiation or AC media (20 ng/ml ActivinA, 3 μM CHIR99021 in N2B27) for mesoderm differentiation. The media were refreshed every 24 h and the plates fixed on day 3 and 7 for immunostaining. Endoderm differentiation was carried out as previously described [72].

### Data deposition

The mass spectrometry proteomics data have been deposited to the ProteomeXchange Consortium [95] via the PRIDE partner repository with the dataset identifier PXD003247.

### Statistical testing

Unpaired Student's *t*-test was used when $n > 2$ for pairwise comparison of mutant or treated cells vs. control. Paired *t*-test was used to compare RGd2 expression in RSK mutant vs. control cells across multiple time points (Fig EV3). Please refer to the phosphoproteomics analysis section for details of the statistical analysis used to identify pERK target proteins.

**Expanded View** for this article is available online.

### Acknowledgements
We are grateful to Jason Wray and Tűzer Kalkan for advice. We thank Andy Riddell for flow cytometry support and Mike Deery for mass spectrometry support. We also thank Rosalind Drummond for valuable technical assistance and Martin Leeb and Jörg Betschinger for helpful discussions. This research was funded by European Commission Projects EuroSyStem (HEALTH-F4-2008-200720) and SyBoss (242129) and the Leverhulme Trust (RPG-2016-418). The Cambridge Stem Cell Institute receives core funding from The Wellcome Trust and The Medical Research Council. AS is a Medical Research Council Professor.

### Author contributions
IREN, KSL and AS designed the study. IREN and CM performed the experiments. IREN, CM and LG analysed the data. IREN, CM and LG prepared the figures. KSL and AS supervised the study. IREN, CM and AS wrote the paper.

### Conflict of interest
The authors declare that they have no conflict of interest.

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
