## [Review Process File · EMBO Reports]

Negative feedback via RSK modulates Erk-dependent progression from naïve pluripotency

Isabelle RE Nett, Carla Mulas, Laurent Gatto, Kathryn S Lilley, Austin Smith

Review timeline:

Submission date:	14 December 2017
Editorial Decision:	23 January 2018
Revision received:	26 April 2018
Editorial Decision:	15 May 2018
Revision received:	16 May 2018
Accepted:	18 May 2018

Editor: Achim Breiling

Transaction Report:

1st Editorial Decision

23 January 2018

Thank you for the submission of your research manuscript to EMBO reports. We have now received reports from the three referees that were asked to evaluate your study, which can be found at the end of this email.

As you will see, all referees support the publication of your paper in EMBO reports. Nevertheless, they all have a number of concerns and/or suggestions to improve the manuscript, which we ask you to address in a revised manuscript. As the reports are below, I will not detail them here. We feel, however, that all points need to be addressed in the revised manuscript. In particular, further evidence needs to be provided that RSK deficient ESCs exit the pluripotent state (point 2 of referee #3), including indications that the data shown in Fig. 4A is reproducible (e.g. by showing a biological replicate, or quantifications of more than one experiment). Moreover, reproducibility and normalization for the data in Fig. 2 needs to be clarified as indicated by the referees (e.g. referee #2 point 3 and referee #3 point 1).

Given the constructive referee comments, we would like to invite you to revise your manuscript with the understanding that all referee concerns must be addressed in the revised manuscript and in a point-by-point response. Acceptance of your manuscript will depend on a positive outcome of a second round of review. It is EMBO reports policy to allow a single round of revision only and acceptance or rejection of the manuscript will therefore depend on the completeness of your responses included in the next, final version of the manuscript.

Revised manuscripts should be submitted within three months of a request for revision; they will otherwise be treated as new submissions. Please contact us if a 3-months time frame is not sufficient for the revisions so that we can discuss the revisions further.

Supplementary/additional data: The Expanded View format, which will be displayed in the main HTML of the paper in a collapsible format, has replaced the Supplementary information. You can submit up to 5 images as Expanded View. Please follow the nomenclature Figure EV1, Figure EV2

etc. The figure legend for these should be included in the main manuscript document file in a section called Expanded View Figure Legends after the main Figure Legends section. Additional Supplementary material should be supplied as a single pdf labeled Appendix. The Appendix includes a table of content on the first page, all figures and their legends. Please follow the nomenclature Appendix Figure Sx throughout the text and also label the figures according to this nomenclature.

For more details (also on the formats "article" and "scientific reports") please refer to our guide to authors:

<http://embor.embopress.org/authorguide#manuscriptpreparation>

Looking at the length of your submission, I think the final manuscript could be published as scientific report (up to 5 main figures and 5 EV figures). In that case, we would ask you to combine the results and discussion into one section called "results and discussion".

See also our guide for figure preparation:

http://www.embopress.org/sites/default/files/EMBOPress_Figure_Guidelines_061115.pdf

Important: All materials and methods should be included in the main manuscript file.

Regarding data quantification and statistics, can you please specify, where applicable, the number "n" for how many experiments were performed, the bars and error bars (e.g. SEM, SD) and the test used to calculate p-values in the respective figure legends. Please provide statistical testing where applicable.

We now strongly encourage the publication of original source data with the aim of making primary data more accessible and transparent to the reader. The source data will be published in a separate source data file online along with the accepted manuscript and will be linked to the relevant figure. If you would like to use this opportunity, please submit the source data (for example scans of entire gels or blots, data points of graphs in an excel sheet, additional images, etc.) of your key experiments together with the revised manuscript. Please include size markers for scans of entire gels, label the scans with figure and panel number, and send one PDF file per figure or per figure panel.

- a complete author checklist, which you can download from our author guidelines (<http://embor.embopress.org/authorguide#revision>). Please insert page numbers in the checklist to indicate where the requested information can be found.
- a letter detailing your responses to the referee comments in Word format (.doc)
- a Microsoft Word file (.doc) of the revised manuscript text
- editable TIFF or EPS-formatted single figure files in high resolution (for main figures and EV figures)

In addition I would need from you:

- a short, two-sentence summary of the manuscript
- two to three bullet points highlighting the key findings of your study
- a schematic summary figure (in jpeg or tiff format with the exact width of 550 pixels and a height of about 400 pixels) that can be used as visual synopsis on our website.

I look forward to seeing a revised version of your manuscript when it is ready. Please let me know if you have questions or comments regarding the revision.

REFEREE REPORTS

Referee #1:

ERK signaling plays a pivotal role in initiating ES cell differentiation. In this manuscript, Nett and

colleagues applied spectrometry-based phosphoproteomics to identify a group of ERK targets in mouse ES cells. Next, they focused on investigating the function of one of the ERK targets, RSK1. They found that disruption of the function of RSKs, mainly RSK1, increased pERK levels and accelerated ES cell differentiation. They further carried out a series experiments to demonstrate that RSK feedback modulates the dynamics of ERK activation and ES cell transition kinetics. In general, this study was well designed and the data are convincing. The data generated in this study will provide a very useful resource for understanding how ES cell fate is regulated by ERK signaling.

Some typewriter errors and other minor comments:

- 1) Line 185 of page 6: a period "." is missing after "pERK levels", and there is an extra of comma between "the" and "small molecule BI-D1870".
- 2) Line 197 of page 7, "gradually declined gradually 2-fold", the extra "gradually" needs to be removed.
- 3) Line 309 of page 10, "examineded" should be "examined".
- 4) Figure 2A, a loading control for the Western blot analysis is missing.
- 5) Figure 2C, why is Western blot analysis for RSK3 not included?
- 6) Figure 3A and 3D, additional quantification data should be presented.
- 7) References 70 and 84 need to be updated

Referee #2:

In their manuscript entitled "RSK feedback on Erk modulates the kinetics of progression from naïve pluripotency", Nett et al. start by identifying proteins phosphorylated by ERK in mouse ES cells that have been released from Mek inhibition. Using through mass spectrometry, they detect a surprisingly small number of only 22 target proteins that show significant changes to their phosphorylation pattern, one of which is the kinase RSK1. Using RSK mutant cell lines and a RSK inhibitor, the authors show that RSKs act as feedback inhibitor of ERK activity in ES cells. Lastly, Nett et al. explore the effects of interfering with the ERK-RSK-feedback and demonstrate that loss of RSK function leads to accelerated loss of naïve pluripotency, likely as a consequence of increased ERK activity.

The data presented in this paper conclusively show that RSK1 is a target and negative feedback regulator of ERK signalling in ESCs that controls the dynamics of exit from naïve pluripotency.

How levels of ERK signalling are controlled and how they regulate transitions between different states of pluripotency is a major unexplored question in the ES cell field, and even though negative feedback regulation of ERK through RSK has been identified in other systems (as appropriately acknowledged by the authors), this paper is the first one to show that this mechanism is important in ESCs and regulates the dynamics of progression from naïve pluripotency. The work thus constitutes an important advance that will of interest to a broad readership. The study is mostly well executed, except for some problems, e.g. with qPCR analysis and genotyping of mutant lines that I describe in more detail below.

Provided that the authors appropriately address these points, I recommend this manuscript for publication in EMBO reports.

Major points:

1. The authors need to discuss in more detail which signalling systems activate ERK in their experimental paradigm. Is ERK activated by factors in the medium or through paracrine ligands? It is possible that this changes depending on when they assay (e.g. after 24h in Fig. 1 vs. after 1h in Fig. 2). Also, it has been shown that LIF can activate ERK (e.g. Ohtsuka et al., 2015), a study that the authors need to discuss in this context (line 114).
2. One of the most surprising findings of the study is the low number of target proteins that are differentially phosphorylated in response to removal of the Mek inhibitor. It is not clear to me whether this is due to low sensitivity of the assay (i.e. few peptides were identified overall), or

because the phosphoproteome did not change between the conditions. To address this question, it would be helpful to mention in the main text how many phosphopeptides and proteins were identified overall, and what the criteria for identification were. At the moment this is touched upon in the methods section, but I think this would be more appropriately discussed in the results section.

3. qPCR experiments in Figs. EV1C, EV2A, EV2D lack error bars. It is also not clear from the methods section how the measurements have been normalized. E.g. I would expect gene expression in one condition to be normalized to 1, but this is not the case in EV1C, EV2A, EV2D, 4B, 5B. The authors should describe their normalization approach more clearly in the methods section, or change it.

4. The initial characterization of the RSK mutant lines is hard to follow (lines 181 onwards, Fig. 2B, 2C), partially because the authors go back and forth between sequence analysis (table EV3) and functional analysis of their cell lines, which are not entirely consistent. E.g. what is the motivation to use BI-D in Fig. 2B? Why do the authors conclude that clone RSK1*23 is heterozygous for RSK2, if they report two alleles that both lead to premature termination in table EV3? The same applies to the state of the RSK4 locus in the RSK3*4 mutant, and the states of the RSK1 and RSK3 loci in the RSK13 mutant. Or are these sequences derived from different clones? The authors need to clarify these issues, and if applicable, also discuss the discrepancy between their sequencing results and the immunoblot in Fig. 2C.

It is also not clear from the methods description how the Sanger sequencing to genotype the different alleles was performed. The standard procedure is to clone PCR amplicons, and then to sequence a number of plasmid clones. If this has been done it should be reported. If not, performing this type of analysis might help to address the basis of the discrepancy between table EV3 and Fig. 2C.

5. Line 299, 300: The data supporting this statement seems to be missing.

Minor points

1. Line 119: Typo, should read "coverage".
2. Boeuf et al. (JBC, 2005) have investigated ERK-mediated phosphorylation of RSK in ESCs before. The authors need to cite this reference.
3. Fig. EV2A could be simplified.
4. A loading control needs to be shown for Fig. 2A.
5. Fig. EV2B needs better explanation - what do the green and black bars represent?
6. Fig. EV 2E: Arrange bottom panel on the side of top two panels, as these are from different blots. Also, probing for total ERK would be informative.
7. Fig. 2D: Please state if and how density measurements have been normalized, as total ERK levels in immunoblot appear to be rather variable between conditions.
8. Line 205: Typo
9. Line 289: Part of sentence missing.
10. Line 371: "The persistence of oscillations": The data supporting this claim is rather weak, the authors may consider to remove this claim.
11. Line 1011: Should read RGd2, not ReGd2

Referee #3:

In this manuscript Nett et al., performed a phosphoproteome screen in naive ESC (2i) upon withdrawal of MEK inhibition, and identify RSK1 phosphorylation as being ERK-dependent. Upon CRISPR deletion of RSK1 (and heterozygous deletion of related family members RSK2 and RSK3) p-ERK levels are elevated upon the first 24 hours of ESC differentiation (withdrawal of 2i). The authors go on to show that the consequence of elevated ERK levels in RSK deficient ESCs is a premature exit from naïve pluripotency. These results can be recapitulated by small molecule inhibition.

The role of ERK signaling in the exit of pluripotency is well documented but the dynamics of regulation of p-ERK is an interesting open question and is of interest to those in the pluripotent stem cell field. While RSKs have been previously shown to be negative regulators of ERK signaling, this study represents the first time it has been shown during ESC differentiation. However, conclusions that RSK depletion influences pERK dynamics are overstated. Moreover, additional data is needed to provide sufficient evidence that the RSK deficient ESCs are prematurely and competently exiting the pluripotent state.

Major Comments:

1. Pg 7 pp2 "We conclude that RSK feedback modulates both the level and the dynamics of ERK activation in ES cells." While the data convincingly demonstrates that ERK levels are consistently elevated upon RSK-depletion, the data in Fig. 2D is not sufficient to support the conclusion that p-ERK dynamics are significantly disrupted. The methods and text / figure legends are not sufficiently detailed on how the quantification of the Western blot was performed. Due to the variable ERK levels, pERK levels should be normalized to total ERK. It also should be stated the sample number for replicates used in the quantification. Moreover, the Western blot appears to be overexposed making it difficult to obtain accurate quantification. It may be informative to look at the transcriptional dynamics of ERK targets such as *Spry2/4*, *Dusp6/9*, *Fos* and *Erg1* to see if the expression kinetics are altered, and if this supports or contradicts conclusions about the altered dynamics of ERK signaling in RSK1*23 mutants.
2. To support the conclusions that RSK deficiency accelerates ESC entry into differentiation and dissolution of pluripotency network, in addition to protein analysis (Fig. 4A) a time-course of RSK1*23 differentiation and qPCR analysis of transcripts of naïve pluripotency markers to see if there is a significant difference to control cells in timing. In addition, if the RSK1*23 cells remain pluripotent it should be demonstrated that these cells can still differentiate towards mesoderm and endoderm lineages, and not just neural differentiation (Fig. 3F).
3. The authors demonstrate with RSK loss of function that pERK levels are elevated, indicating that RSKs negatively regulates pERK levels to allow for a timely exit of pluripotency. Can the authors comment as to why in the rescue experiments when RSK1 is overexpressed in the control + pBSK1 (Extended Fig. 2E) there is not a decrease in pERK relative to the control?
4. P10 pp 3 "BI-treated cells acquired a GFP low population more synchronously (Fig 5C) ..." It is not immediately clear to me from the flow cytometry time-course support that this is a more synchronous process, but it just happens more quickly, and would greatly benefit from quantification to show if there is a significant reduction in heterogeneity at equivalent % of Rex1GFP+ cells.

Minor comments:

1. Abstract states "Mitogen-activated protein kinase (ERK) signaling" Do the authors mean (MAPK) / ERK signaling or Extracellular and related kinase (ERK).
2. Explain what RGd2 abbreviation/cells are, it is not immediately clear from the text.
3. When discussing the timed treatments in Fig. EV4 it should be stated in that the exit from pluripotency is assessed by Rex1 GFP+ expression.
4. Fig. 5D is % of Rex1GFP+ at 24hrs, but the flow cytometry histograms for a 24hr time point are not shown.
5. P9 final pp. "... With a peak at 30 min followed by gradual." Sentence ends abruptly.

Referee #1:

ERK signaling plays a pivotal role in initiating ES cell differentiation. In this manuscript, Nett and colleagues applied spectrometry-based phosphoproteomics to identify a group of ERK targets in mouse ES cells. Next, they focused on investigating the function of one of the ERK targets, RSK1. They found that disruption of the function of RSKs, mainly RSK1, increased pERK levels and accelerated ES cell differentiation. They further carried out a series of experiments to demonstrate that RSK feedback modulates the dynamics of ERK activation and ES cell transition kinetics. In general, this study was well designed and the data are convincing. The data generated in this study will provide a very useful resource for understanding how ES cell fate is regulated by ERK signaling.

Some typewriter errors and other minor comments:

1) Line 185 of page 6: a period "." is missing after "pERK levels", and there is an extra comma between "the" and "small molecule BI-D1870".
Amended.

2) Line 197 of page 7, "gradually declined gradually 2-fold", the extra "gradually" needs to be removed.
Amended.

3) Line 309 of page 10, "examineded" should be "examined".
Amended.

4) Figure 2A, a loading control for the Western blot analysis is missing.
Added.

5) Figure 2C, why is Western blot analysis for RSK3 not included?
Unfortunately, we could not find any specific antibody for RSK3. In order to determine the expression of RSK3 in the RSK1*23 clone, we carried out RT-qPCR (Fig EV2D).

6) Figure 3A and 3D, additional quantification data should be presented.
Quantification across experiments and timepoints has been added (Fig EV3A).

7) References 70 and 84 need to be updated.
Unfortunately we are not really sure in what way these references need updating.

Referee #2:

In their manuscript entitled "RSK feedback on Erk modulates the kinetics of progression from naïve pluripotency", Nett et al. start by identifying proteins phosphorylated by ERK in mouse ES cells that have been released from Mek inhibition. Using through mass spectrometry, they detect a surprisingly small number of only 22 target proteins that show significant changes to their phosphorylation pattern, one of which is the kinase RSK1. Using RSK mutant cell lines and a RSK inhibitor, the authors show that RSKs act as feedback inhibitor of ERK activity in ES cells. Lastly, Nett et al. explore the effects of interfering with the ERK-RSK-feedback and demonstrate that loss of RSK function leads to accelerated loss of naïve pluripotency, likely as a consequence of increased ERK activity.

The data presented in this paper conclusively show that RSK1 is a target and negative feedback regulator of ERK signalling in ESCs that controls the dynamics of exit from naïve pluripotency.

How levels of ERK signalling are controlled and how they regulate transitions between different states of pluripotency is a major unexplored question in the ES cell field, and even though negative feedback regulation of ERK through RSK has been identified in other systems (as appropriately acknowledged by the authors), this paper is the first one to show that this mechanism is important in ESCs and regulates the dynamics of progression from naïve pluripotency. The work thus constitutes an important advance that will be of interest to a broad readership. The study is mostly well executed, except for some problems, e.g. with qPCR analysis and genotyping of mutant lines that I describe in more detail below.

Provided that the authors appropriately address these points, I recommend this manuscript for

publication in EMBO reports.

Major points:

1. The authors need to discuss in more detail which signalling systems activate ERK in their experimental paradigm. Is ERK activated by factors in the medium or through paracrine ligands? It is possible that this changes depending on when they assay (e.g. after 24h in Fig. 1 vs. after 1h in Fig. 2). Also, it has been shown that LIF can activate ERK (e.g. Ohtsuka et al., 2015), a study that the authors need to discuss in this context (line 114).

It is well-established that autocrine FGF4 is the primary activator of the ERK1/2 pathway in serum-free ES cell cultures, although insulin in the medium may also contribute. This is stated in both the introduction and in the first sentence of results, with relevant citations.

2. One of the most surprising findings of the study is the low number of target proteins that are differentially phosphorylated in response to removal of the Mek inhibitor. It is not clear to me whether this is due to low sensitivity of the assay (i.e. few peptides were identified overall), or because the phosphoproteome did not change between the conditions. To address this question, it would be helpful to mention in the main text how many phosphopeptides and proteins were identified overall, and what the criteria for identification were. At the moment this is touched upon in the methods section, but I think this would be more appropriately discussed in the results section. The description of the analysis settings used for quantification of the phosphopeptides has now been added to the result section. The low number of target proteins identified is unexpected but may be related to the naïve state of ES cells under the conditions of assay (24 hrs in CH+LIF). In future studies it will be of interest to identify ERK targets in ES cells undergoing transition.

3. qPCR experiments in Figs. EV1C, EV2A, EV2D lack error bars. It is also not clear from the methods section how the measurements have been normalized. E.g. I would expect gene expression in one condition to be normalized to 1, but this is not the case in EV1C, EV2A, EV2D, 4B, 5B. The authors should describe their normalization approach more clearly in the methods section, or change it.

For siRNA and rescue experiments, RT-qPCR data are relative to GAPDH and normalised to control siRNA or untransfected RGD2 cells respectively. For RT-qPCR during differentiation, normalisation to a single sample was not deemed appropriate. Therefore, the expression values are shown relative to *Gapdh* or *Actb* (no normalisation). The error bars have been added and an explanation of the normalisation has been added to the legend. The axes have been amended to make the data processing clear.

4. The initial characterization of the RSK mutant lines is hard to follow (lines 181 onwards, Fig. 2B, 2C), partially because the authors go back and forth between sequence analysis (table EV3) and functional analysis of their cell lines, which are not entirely consistent. E.g. what is the motivation to use BI-D in Fig. 2B?

We have added a sentence to clarify the use of BI. The aim was to gauge the extent to which RSK function was still present in the different RSK mutants.

Why do the authors conclude that clone RSK1*23 is heterozygous for RSK2, if they report two alleles that both lead to premature termination in table EV3?

In Table EV3 (now Table EV1) we reported that a single RSK2 allele (*Rps6ka3*) is mutated. Genomic PCR detected two truncated alleles for RSK3 (*Rps6ka2*). Unfortunately, we could not find any specific antibody for RSK3. RT-qPCR for *Rps6ka2* (RSK3), showed mRNA levels similar to wild type (Fig EV2D - added). We concluded that the cell line is mutant for RSK3 but may still produce some protein. We have clarified this in the text.

The same applies to the state of the RSK4 locus in the RSK3*4 mutant, and the states of the RSK1 and RSK3 loci in the RSK13 mutant. Or are these sequences derived from different clones? The authors need to clarify these issues, and if applicable, also discuss the discrepancy between their sequencing results and the immunoblot in Fig. 2C.

In the RSK3*4 mutant, one RSK4 allele carries a 3 bp deletion (Table EV3 – now table EV1) and the protein levels appear unaltered by western blot (Fig. 2C). Therefore, we conclude that the 3bp deletion caused a single amino acid deletion that did not alter the RSK4 protein levels and whose effect on RSK4 activity is unknown. We labelled this clone having possible residual RSK4 activity.

We have added a new table (new Table EV3), summarising all the information regarding the genotype of each RSK clone.

It is also not clear from the methods description how the Sanger sequencing to genotype the different alleles was performed. The standard procedure is to clone PCR amplicons, and then to sequence a number of plasmid clones. If this has been done it should be reported. If not, performing this type of analysis might help to address the basis of the discrepancy between table EV3 and Fig.

2C.

Four to eight cloned PCR amplicons were sequenced for each RSK clone. We have added a statement explaining this in the methods.

5. Line 299, 300: The data supporting this statement seems to be missing.

The statement "RSK1 is the major effector but can be compensated by combined activity of both RSK2 and RSK3, though not by either alone." refers to changes in pERK levels as a result of siRNA knockdown of different RSK isoforms. A change in pERK was only observed in RSK1, 2 and 3 triple knockdown, but not in double RSK1 and 2 or RSK1 and 3 knockdown. Furthermore, only conditions with RSK1 siRNA upregulated pERK, which suggest RSK1 is the major effector.

Minor points

1. Line 119: Typo, should read "coverage".
Amended.

2. Boeuf et al. (JBC, 2005) have investigated ERK-mediated phosphorylation of RSK in ESCs before. The authors need to cite this reference.
Possibly the reviewer means Boeuf et al, 2001? The reference has been added.

3. Fig. EV2A could be simplified.
Amended.

4. A loading control needs to be shown for Fig. 2A.
Added.

5. Fig. EV2B needs better explanation - what do the green and black bars represent?
We have added explanations of the diagrams both in the figure and caption.

6. Fig. EV 2E: Arrange bottom panel on the side of top two panels, as these are from different blots. Also, probing for total ERK would be informative.
We have rearranged the figure and increased the separation between the panels to emphasise they are from different blots. In previous experiments, modulation of RSK levels did not change total ERK levels (Fig 2B). Therefore, we do not feel this control would add further information than the tubulin staining (loading control).

7. Fig. 2D: Please state if and how density measurements have been normalized, as total ERK levels in immunoblot appear to be rather variable between conditions.
We have repeated the analysis and quantified the graph as the ratio of the pERK/ERK intensity. To quantify the blots, we used imageJ to draw boxes of equal size around the bands and calculated mean intensity. A description of the quantification has been added to the methods. We have also provided the biological replicate in the Fig EV2.

8. Line 205: Typo
Amended.

9. Line 289: Part of sentence missing.
Amended.

10. Line 371: "The persistence of oscillations": The data supporting this claim is rather weak, the authors may consider to remove this claim.
Amended.

11. Line 1011: Should read RGd2, not ReGd2
Amended.

Referee #3:

In this manuscript Nett et al., performed a phosphoproteome screen in naive ESC (2i) upon withdrawal of MEK inhibition, and identify RSK1 phosphorylation as being ERK-dependent. Upon CRISPR deletion of RSK1 (and heterozygous deletion of related family members RSK2 and RSK3) p-ERK levels are elevated upon the first 24 hours of ESC differentiation (withdrawal of 2i). The authors go on to show that the consequence of elevated ERK levels in RSK deficient ESCs is a premature exit from naïve pluripotency. These results can be recapitulated by small molecule

inhibition.

The role of ERK signaling in the exit of pluripotency is well documented but the dynamics of regulation of p-ERK is an interesting open question and is of interest to those in the pluripotent stem cell field. While RSKs have been previously shown to be negative regulators of ERK signaling, this study represents the first time it has been shown during ESC differentiation. However, conclusions that RSK depletion influences pERK dynamics are overstated. Moreover, additional data is needed to provide sufficient evidence that the RSK deficient ESCs are prematurely and competently exiting the pluripotent state.

Major Comments:

1. Pg 7 pp2 "We conclude that RSK feedback modulates both the level and the dynamics of ERK activation in ES cells." While the data convincingly demonstrates that ERK levels are consistently elevated upon RSK-depletion, the data in Fig. 2D is not sufficient to support the conclusion that p-ERK dynamics are significantly disrupted. The methods and text / figure legends are not sufficiently detailed on how the quantification of the Western blot was performed. Due to the variable ERK levels, pERK levels should be normalized to total ERK. It also should be stated the sample number for replicates used in the quantification.

Moreover, the Western blot appears to be overexposed making it difficult to obtain accurate quantification.

We agree with the reviewer that our population data do not allow a firm conclusion that dynamics are substantially altered upon RSK deletion and we have removed these comments. We have now included in Fig EV2G the biological duplicate of the Western blot timecourse and re-analysed the data to provide the intensity quantification as a ratio of pERK/ERK.

It may be informative to look at the transcriptional dynamics of ERK targets such as *Spry2/4*, *Dusp6/9*, *Fos* and *Erg1* to see if the expression kinetics are altered, and if this supports or contradicts conclusions about the altered dynamics of ERK signaling in RSK1*23 mutants.

We analysed the expression of pERK targets and this showed that RSK1*23 mutants show altered expression dynamics for some targets. These new data have been added to the manuscript (Fig EV2H).

2. To support the conclusions that RSK deficiency accelerates ESC entry into differentiation and dissolution of pluripotency network, in addition to protein analysis (Fig. 4A) a time-course of RSK1*23 differentiation and qPCR analysis of transcripts of naïve pluripotency markers to see if there is a significant difference to control cells in timing.

We took advantage of the BI inhibition system to examine the relationship between mRNA and protein down-regulation for pluripotency factors. These new data, which complement the results in Fig 4A, are presented in Fig. EV4B,C.

In addition, if the RSK1*23 cells remain pluripotent it should be demonstrated that these cells can still differentiate towards mesoderm and endoderm lineages, and not just neural differentiation (Fig. 3F).

We have added immunostaining analyses showing RSK1*23 cell differentiation towards all three germ layers (Fig 3 F-H).

3. The authors demonstrate with RSK loss of function that pERK levels are elevated, indicating that RSKs negatively regulates pERK levels to allow for a timely exit of pluripotency. Can the authors comment as to why in the rescue experiments when RSK1 is overexpressed in the control + pBSK1 (Extended Fig. 2E) there is not a decrease in pERK relative to the control?

RSK has to be phosphorylated by pERK to become activated and exert its negative regulatory function. Therefore, pERK is limiting, rather than total RSK levels (as also indicated by the requirement for compound depletion of RSKs).

4. P10 pp 3 "BI-treated cells acquired a GFP low population more synchronously (Fig 5C) ..." It is not immediately clear to me from the flow cytometry time-course support that this is a more synchronous process, but it just happens more quickly, and would greatly benefit from quantification to show if there is a significant reduction in heterogeneity at equivalent % of Rex1GFP+ cells.

To address this, we have plotted the mean RGd2 fluorescence levels against the coefficient of variation for all experimental data points (25h BI or DMSO treatment, analysed at different timepoints) – Fig EV5A. For every mean fluorescence value, the coefficient of variation was lower in BI-treated cells. To avoid ambiguity we now refer to the process as more uniform rather than more synchronous.

Minor comments:

1. Abstract states "Mitogen-activated protein kinase (ERK) signaling" Do the authors mean (MAPK) / ERK signaling or Extracellular and related kinase (ERK).
Amended.
2. Explain what RGd2 abbreviation/cells are, it is not immediately clear from the text.
We have added an explanatory paragraph in the methods section describing the cell line (under 'ES cell culture').
3. When discussing the timed treatments in Fig. EV4 it should be stated in that the exit from pluripotency is assessed by Rex1 GFP+ expression.
Amended.
4. Fig. 5D is % of Rex1GFP+ at 24hrs, but the flow cytometry histograms for a 24hr time point are not shown.
This was a labelling mistake from different file versions. All experiments were performed 25hrs after removal of 2i. We have amended this in the text and figure.
5. P9 final pp. "... With a peak at 30 min followed by gradual." Sentence ends abruptly.
Amended.

2nd Editorial Decision

15 May 2018

Thank you for the submission of your revised manuscript to our editorial offices. We have now received the reports from the two referees that were asked to re-evaluate your study (you will find enclosed below). As you will see, both referees now support the publication of your manuscript in EMBO reports. However, referee #1 noted some further minor issues we ask you to address in a final revised manuscript.

Further, I have the following editorial requests:

The title reads rather complicated. Maybe, we could agree on a different one. I would suggest:
Negative feedback via RSK modulates Erk-dependent exit from naïve pluripotency

Regarding data quantification and statistics, can you please specify the number "n" for how many independent experiments (biological replicates) were performed in all the figure legends, add error bars (e.g. SEM, SD) and indicate the test used to calculate p-values. Please add a paragraph about the statistical testing used throughout the manuscript to the method section. Could statistical testing also be provided for the diagrams shown in Figures EV2A/D/E and EV4D/E (provided more than two replicates are shown)?

For Figure 2D you indicate that two replicates have been performed. Could the quantification of the second replicate be shown in the diagram at the bottom (both replicates with separate data points)?

Please add scale bars to the microscopic images shown in Figure EV1.

For the Western Blots shown in Figure 4A and EV2G it seems the cropped images were pasted on top of a grey background, which looks a bit odd. Could you provide the panels with the black frames directly fit to the original images, without the grey background (like in the other Western blot panels)?

As all Western blots have been significantly cropped, we would ask you to submit the source data for the Western blots with the final revised manuscript. The source data will be published in a separate source data file online along with the accepted manuscript and will be linked to the relevant figure. Please submit the source data (scans of entire blots) of all Western blots (main figures and EV figures) together with the revised manuscript. Please include size markers for scans of entire gels, label the scans with figure and panel number, and send one PDF file per figure.

Finally, please find attached a word file of the manuscript text (provided by our publisher) with changes we ask you to include in your final manuscript text, and some queries (comments), we ask you to address. Please provide your final manuscript file with track changes, in order that we can see the modifications done.

When submitting your revised manuscript, we will require:
- a Microsoft Word file (.doc) of the revised manuscript text

- editable TIFF or EPS-formatted figure files (main figures and EV figures) in high resolution (of those with adjusted panels or labels).
- source data for Western blots

I look forward to seeing the final revised version of your manuscript when it is ready. Please let me know if you have questions or comments regarding the revision.

REFEREE REPORTS

 Referee #2:

All my comments from the previous round of review have been satisfactorily addressed. This is a fine manuscript that will be of general interest to scientists interested in the regulation of FGF/ERK signalling and its relationship to cell differentiation.

Minor issues that could be changed in the published version:

P7, line 213: The authors appear to refer to Fig. EV2H, not EV3H.

P13, line 410: ..."Nonetheless ,RSK" exchange comma and space.

Fig. 4B: Color code for control and RSK1*23 lines is reversed compared to all other figures.

 Referee #3:

The authors have satisfactorily addressed my concerns and comments in their revised manuscript.

2nd Revision - authors' response

16 May 2018

The title reads rather complicated. Maybe, we could agree on a different one. I would suggest: Negative feedback via RSK modulates Erk-dependent exit from naïve pluripotency
 This is reasonable, if exit can be changed to progression: Negative feedback via RSK modulates Erk-dependent progression from naïve pluripotency

Regarding data quantification and statistics, can you please specify the number "n" for how many independent experiments (biological replicates) were performed in all the figure legends, add error bars (e.g. SEM, SD) and indicate the test used to calculate p-values. Please add a paragraph about the statistical testing used throughout the manuscript to the method section. Could statistical testing also be provided for the diagrams shown in Figures EV2A/D/E and EV4D/E (provided more than two replicates are shown)?

We have now provided information on all numbers of biological replicates as well as the type of error bar and the statistical test conducted (for n>2). We also added a brief description of the statistical test used in the methods.

For Figure 2D you indicate that two replicates have been performed. Could the quantification of the second replicate be shown in the diagram at the bottom (both replicates with separate data points)?

The replicate is shown in Fig EV2G. The difference in quantitative signal between the two experiments makes it problematic to plot both on the same diagram.

Please add scale bars to the microscopic images shown in Figure EV1.

Unfortunately, the metadata for the original images have been lost. We have indicated in the caption that the images represent a field of view under 20X magnification.

For the Western Blots shown in Figure 4A and EV2G it seems the cropped images were pasted on top of a grey background, which looks a bit odd. Could you provide the panels with the black frames directly fit to the original images, without the grey background (like in the other Western blot panels)?

We changed the background of Figure 4A and EV2G to black as requested.

As all Western blots have been significantly cropped, we would ask you to submit the source data for the Western blots with the final revised manuscript. The source data will be published in a separate source data file online along with the accepted manuscript and will be linked to the relevant figure. Please submit the source data (scans of entire blots) of all Western blots (main figures and EV figures) together with the revised manuscript. Please include size markers for scans of entire gels, label the scans with figure and panel number, and send one PDF file per figure.

Finally, please find attached a word file of the manuscript text (provided by our publisher) with changes we ask you to include in your final manuscript text, and some queries (comments), we ask you to address. Please provide your final manuscript file with track changes, in order that we can see the modifications done.

We have added all the relevant information.

- a Microsoft Word file (.doc) of the revised manuscript text
- editable TIFF or EPS-formatted figure files (main figures and EV figures) in high resolution (of those with adjusted panels or labels).
- source data for Western blots

These have been uploaded.

Referee #2:

All my comments from the previous round of review have been satisfactorily addressed. This is a fine manuscript that will be of general interest to scientists interested in the regulation of FGF/ERK signalling and its relationship to cell differentiation.

Minor issues that could be changed in the published version:

P7, line 213: The authors appear to refer to Fig. EV2H, not EV3H.
Corrected.

P13, line 410: ..."Nonetheless ,RSK" exchange comma and space.
Corrected.

*Fig. 4B: Color code for control and RSK1*23 lines is reversed compared to all other figures.*
Corrected.

Referee #3:

The authors have satisfactorily addressed my concerns and comments in their revised manuscript.

Corresponding Author Name: Austin Smith

Manuscript Number: EMBOR-2017-45642V2-Q